# Association of the Vitamin D Level and Quality of School Life in Adolescents with Irritable Bowel Syndrome

**DOI:** 10.3390/jcm7120500

**Published:** 2018-12-01

**Authors:** Youngsun Cho, Yoomi Lee, Youjin Choi, Sujin Jeong

**Affiliations:** 1Department of Pediatrics, CHA Bundang Medical Center, CHA University School of Medicine, Seongnam 13496, Korea; chocolo@naver.com (Y.C.); lymangel@chamc.co.kr (Y.L.); 2Department of Pediatrics, Inje University, Ilsan-Paik Hospital, Goyang 10380, Korea; jaenia83@gmail.com

**Keywords:** irritable bowel syndrome, functional gastrointestinal disorder, vitamin D supplementation, school aged children, Korea, quality of school life

## Abstract

There is no treatment of choice for irritable bowel syndrome, which affects up to 20% of school-aged children. This cross-sectional study evaluated the difference in the average vitamin D level between subtypes of irritable bowel syndrome, and the relationship between the vitamin D level as well as the severity of irritable bowel syndrome symptoms. We included 124 adolescents aged 10–17 years (68 boys, 56 girls; mean age 12.29 ± 1.92 years) from 2014 to 2016. Patients with irritable bowel syndrome were diagnosed by Rome III criteria and classified by clinical manifestation: irritable bowel syndrome with constipation (*n* = 29), irritable bowel syndrome with diarrhea (*n* = 63), and irritable bowel syndrome with constipation and diarrhea (*n* = 32). The severity of irritable bowel syndrome symptoms and school absence were evaluated. Vitamin D levels were measured by serum 25-hydroxyvitamin D. The chi-square test and analysis of variance were used. The patients’ average vitamin D level was 16.25 ± 6.58 ng/mL. There was a significant negative association of the 25-hydroxyvitamin D level with symptom severity and school absence (*p* = 0.022 and *p* < 0.001, respectively). Vitamin D supplementation could be considered as a choice of therapeutic method.

## 1. Introduction

Functional gastrointestinal disorders (FGIDs) are characterized by recurring gastrointestinal (GI) symptoms that cannot be attributed to structural or biochemical abnormalities [1]. Childhood irritable bowel syndrome (IBS), one of 10 FGIDs according to the ROME III diagnostic criteria, is a common FGID affecting up to 20% of school-aged children [2,3]. IBS is a chronic condition characterized by abdominal discomfort or pain associated with altered bowel habits and flatulence, with various symptom severities, ranging from mild to severe. IBS symptoms are associated with psychological distress, leading to impaired social and personal functions and to a decline in the patient’s quality of life [4]. Concomitantly, recurrent symptoms, present throughout adolescence and persisting in adulthood, have a major impact on health and social costs [1,5].

However, there is no treatment of choice for IBS. Therapy for IBS is primarily targeted at relieving symptoms; yet, approximately one-third of patients with IBS fail to respond to existing treatments, and 16–33% of outpatients seek alternative medicine [1,5,6,7].

Vitamin D has long been recognized as a major regulator of calcium and phosphorus metabolism. The rate of vitamin D deficiency has been estimated to be 30–50% worldwide, which is also a common percentage of patients with GI disease [8,9,10,11,12,13,14].

Although the role of vitamin D in IBS has not been determined, a recent study reported the successful treatment of diarrhea-predominant IBS with high doses of oral vitamin D supplementation [15]. Furthermore, an analysis of the social media posts (blogs/forums) of 37 adult patients with IBS, who reported themselves as being vitamin D deficient, indicated that 70% described improvements in their symptoms after using high-dose vitamin D supplementation [15]. On the basis of this research, we conducted this study to evaluate the difference in the average vitamin D level between three subtypes of pediatric patients with IBS and to determine the relationship between vitamin D level and IBS symptoms.

## 2. Experimental Section

This cross-sectional study was conducted from 2014 to 2016. Participants with IBS symptoms were recruited from a pediatric gastroenterology clinic at CHA Bundang Medical Center. The exclusion criteria for this study was the presence of chronic liver disease, malabsorption syndrome, inflammatory bowel disease, diseases affecting calcium and/or vitamin D metabolism and any treatment with steroids, vitamin D, or calcium supplementation before the date of 25-hydroxyvitamin D (25-OHD) measurement. Of the 129 subjects with IBS who met the inclusion criteria, five were excluded based on the aforementioned exclusion criteria.

Patients with IBS were diagnosed by the ROME III criteria and classified as having three types of IBS according to their clinical manifestations using the Bristol stool form scale: IBS with constipation (IBS-C, *n* = 29), IBS with diarrhea (IBS-D, *n* = 63), and IBS with constipation and diarrhea (IBS-M, *n* = 32).

Demographic data including age, sex, and body parameters were collected through a medical record review. The serum concentration of vitamin D was measured by 25-OHD levels, utilizing chemiluminescent assay. We considered vitamin D deficiency as a serum 25-OHD level <20 ng/mL. A serum 25-OHD level exceeding 20 ng/mL was considered as vitamin D sufficiency.

IBS symptom severity and school absence due to IBS symptoms were evaluated with a questionnaire. The questionnaire consisted of the Bristol stool scale and four questions about symptoms and their effect on the patient’s quality of school life (Appendix A).

IBS subtyping was based on the Bristol stool form scale, which is a validated stool form scale that classifies human feces into seven categories based on stool cohesion and surface cracking. Patients with types I and II stools are considered to have constipation (hard stool), and those with types VI and VII are considered to have diarrhea (loose stool).

Statistical analyses were performed using SPSS 23 (IBM Corp., Armonk, NY, USA). Quantitative data are presented as means ± standard deviations. Qualitative values are presented as numbers and percentages.

The chi-square test was used to assess the distribution of participants by vitamin D stratification. Associations between symptom severity and 25-OHD were determined by analysis of variance with Bonferroni corrections. *p*-values < 0.05 were considered statistically significant.

Subsequently, the association between vitamin D deficiency and abdominal pain or quality of school life was examined by the linear-by-linear association test. 

This study was approved by the institution’s ethical committee (approval number 2015-196), and the study protocol conforms to the Ethical Guidelines of the 1975 Declaration of Helsinki (6th revision, 2008). Written informed consent was obtained from the parents or guardians of the participating children.

## 3. Results

The baseline characteristics of the 124 enrolled patients are shown in Table 1. These children fulfilled the pediatric Rome III criteria for IBS (68 boys (54.8%), 56 girls (45.2%); mean age 12.29 ± 1.92 (range 10–17) years). There was no statistically significant difference in age, sex, and body mass index (BMI) between the three types of IBS. There was also no meaningful seasonal variation in vitamin D levels.

The mean 25-OHD level was 16.25 ± 6.58 ng/mL. The average vitamin D level of the IBS-D patients (15.25 ± 7.25 ng/mL) was lower than that of the other types (18.47 ± 7.37 ng/mL in IBS-C, 16.22 ± 3.29 ng/mL in IBS-M), but the difference was not statistically significant (*p* = 0.091).

There was a marked negative association between the frequency of abdominal pain and the 25-OHD level. Patients who reported more frequent abdominal pain or abdominal bloating showed a lower 25-OHD level (Figure 1). The average 25-OHD level of patients who reported abdominal discomfort everyday was significantly lower than that of those who complained of abdominal pain once per week (14.29 ± 6.89 ng/mL versus (vs.) 22.21 ± 8.35 ng/mL, *p* = 0.022).

Regardless of the type of IBS, the frequency of defecation was not related to the average 25-OHD level. The mean 25-OHD level was related to symptom relief after defecation (Figure 2). The 25-OHD level of the group whose symptoms were relieved to some degree after defecation was significantly lower than that of those whose symptoms were completely relieved after defecation (12.70 ± 5.27 ng/mL vs. 18.24 ± 7.13 ng/mL, *p* = 0.012). The 25-OHD level of those whose symptoms were relieved to some degree after defecation was also significantly lower than that of those whose symptoms were notably relieved after defecation (12.70 ± 5.27 ng/mL vs. 16.51 ± 5.88 ng/mL, *p* = 0.009).

Most importantly, there was a significant negative association between 25-OHD and the frequency of school absence due to IBS symptoms (*p* < 0.001) (Figure 3). Students who were absent from school because of IBS symptoms within the two weeks preceding the study showed significantly lower vitamin D levels than students having no such difficulties in school life (10.53 ± 3.98 ng/mL vs. 21.49 ± 5.86 ng/mL, *p* < 0.001). Moreover, the average vitamin D level of students who left during school hours or visited the health office because of IBS symptoms within the two weeks preceding the study was significantly lower than that of those who occasionally complained about IBS symptoms but had no marked difficulties in school life (12.92 ± 5.20 ng/mL vs. 17.91 ± 5.33 ng/mL, *p* = 0.002).

Among the 124 patients, 88 (70.4%) had a vitamin D level <20 ng/mL (vitamin D deficiency group). There was no significant difference between the vitamin D deficiency and sufficiency groups in age, sex, and BMI. According to the linear-by-linear association analysis, the vitamin D deficiency group experienced more abdominal pain (*p* = 0.03) and was absent more often from school (*p* < 0.01) than the vitamin D sufficiency group. More patients in the vitamin D deficiency group than in the vitamin D sufficiency group complained of daily abdominal pain. Moreover, more patients in the vitamin D deficiency group than in the vitamin D sufficiency group were absent from school (Table 2).

## 4. Discussion

We revealed three important results. Firstly, the average vitamin D level was low in adolescents with IBS (16.25 ± 6.58 ng/mL). Secondly, there was a significant negative correlation between the symptom severity and vitamin D level. Thirdly, there was a significant negative correlation between school absence and the vitamin D level.

Although there are several reasons why patients with IBS show a low serum vitamin D considering that diet and outdoor activity are restricted for most of them, the association between the vitamin D level and pathogenesis of IBS has been proven in other reports [16,17]. Many studies regarding IBS pathophysiology have revealed several causative factors such as colonic dysmotility, visceral hypersensitivity, and brain–gut interaction. Recently, inflammation, especially post-infectious low-grade inflammation, immunologic factors, altered microbiota, dietary factors, and enteroendocrine cells have been implicated [18,19,20].

Vitamin D is known to be a potential immunomodulatory and anti-inflammatory factor, in addition to its classic function as a major regulator of calcium and phosphorus metabolism.

It should also be noted that the gut, rich in microflora, acts as an excellent region for the activation of an immune response, promoting the effect of type-1 helper T cells [21]. Moreover, some studies have provided increasing evidence that the vitamin D pathway is a potentially important modifier of the effects of microbiota on gut inflammation [22].

In this study, we showed that more frequent abdominal pain was associated with a lower vitamin D level. In patients with IBS, the inflammatory response has been known to play an important role in the induction of altered colonic physiology generating the IBS symptoms. In addition, inflammation leads to a heightened nervous system sensitivity, which causes visceral hypersensitivity and abdominal pain perception [23,24,25,26].

The binding of the 25-OHD-vitamin D receptor complex results in the expression of 1-α-hydroxylase, which converts 25-OHD to 1,25-dihydroxyvitamin D. This 25-OHD metabolite has been shown to regulate epithelial barrier function and bowel inflammation, suggesting that a vitamin D-deficiency may directly affect bowel function and IBS symptomology [27,28].

Crucially, the vitamin D level showed a significant negative association with the frequency of absence from school. A large number of our patients with IBS in the vitamin D deficiency group had a difficult school life. Recent studies have shown that the serum 25-OHD level in most adult patients with IBS was <20 ng/mL, and they demonstrated an improvement in the quality of life of adult patients with IBS who received vitamin D supplementation [29,30,31]. In pediatric patients with IBS, school performance is considered as an indicator of quality of life. Therefore, the quality of school life of pediatric IBS patients with a low vitamin D level was worse than that of those with a high vitamin D level.

The vitamin D receptor is expressed in the gut and throughout the nervous system; it regulates neurotransmitter levels, serotonin synthesis and upregulates neurotrophins. Thus, vitamin D may directly affect gut function, neurological development, and IBS symptoms [26,32,33,34,35,36]. Additionally, vitamin D could improve psychological distress and low-grade mucosal inflammation that can affect the sensory–motor system of the gut, alter gut function, and cause the visceral hypersensitivity responsible for IBS symptoms [37,38,39,40].

This study was limited by the fact that vitamin D balance was dependent on numerous physiological and social factors such as exposure to sunlight, physical activity, and dietary habits. Another limitation was that the questionnaire used in this study was not universally validated. As there was no official validated questionnaire, it was more difficult to evaluate pediatric or adolescent quality of life than that of adults. Therefore, the questionnaire used in this study was created based on the questionnaire used in a study on adults [31,41,42]. Moreover, adolescent quality of life was closely associated with school life as adolescents spend most of their time at school.

Fecal calprotectin is an objective and non-invasive sensitive inflammatory marker related to various pathological conditions. Therefore, further evaluation of the relationship between fecal calprotectin and irritable bowel syndrome symptom severity, especially quality of life, is needed.

Based on these results, close monitoring of the vitamin D status of pediatric patients with IBS might provide a clue for predicting their symptom severity and school performance. Furthermore, it is necessary to achieve a proper vitamin D status by increasing outdoor activity and the consumption of dairy products, and vitamin D supplementation for pediatric patients with IBS—this could improve their symptoms and quality of school life.

Our study also showed a significant association between the vitamin D level and symptoms, as well as the quality of school life through linear-by-linear association analysis. The average vitamin D level of adolescents with IBS in this study was <20 ng/mL, and they experienced difficulties in their school life. Further longitudinal research is needed to establish the therapeutic role of vitamin D in the management of pediatric IBS patients. In particular, it is necessary to determine a quantifiable amount and proper duration of vitamin D supplementation in order to establish the optimal dose–response effects.

## Figures and Tables

**Figure 1 jcm-07-00500-f001:**
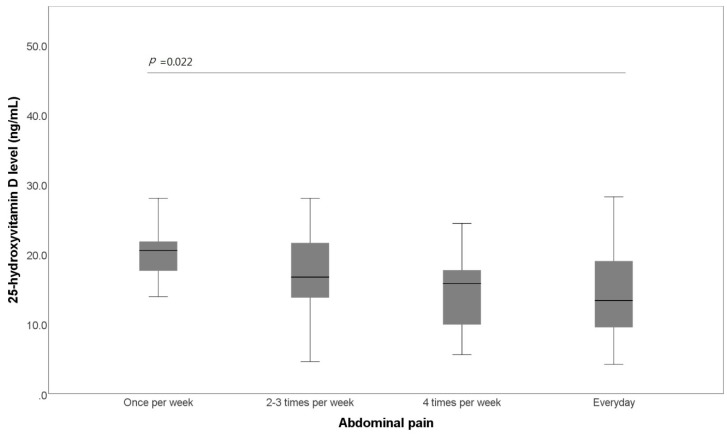
Box plot of the comparison of the 25-hydroxyvitamin D level and abdominal pain. Patients with irritable bowel syndrome are stratified by the frequency of abdominal pain.

**Figure 2 jcm-07-00500-f002:**
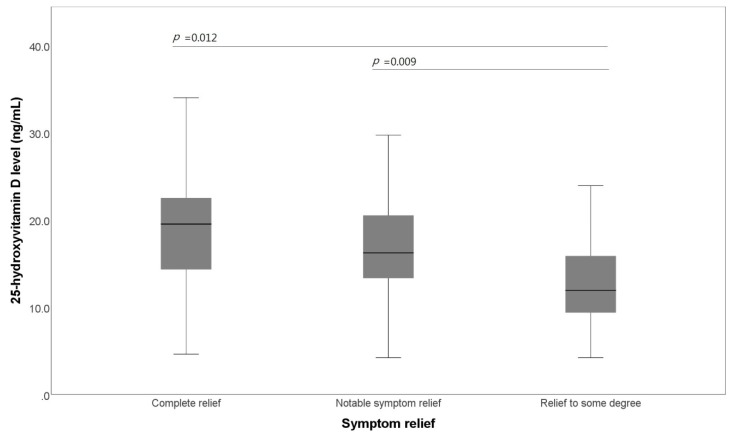
Box plot of the comparison of the 25-hydroxyvitamin D level and symptom relief. Patients with irritable bowel syndrome are stratified by the degree of symptom relief.

**Figure 3 jcm-07-00500-f003:**
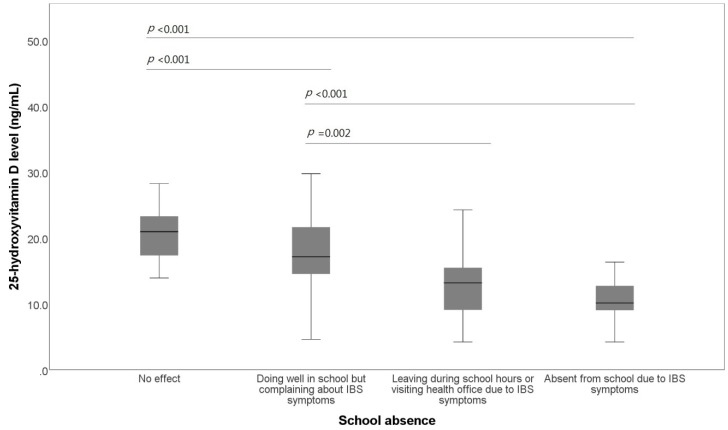
Box plot of the comparison of the 25-hydroxyvitamin D level and school absence. Patients with irritable bowel syndrome are stratified by the frequency of school absence. IBS, irritable bowel syndrome.

**Table 1 jcm-07-00500-t001:** Baseline characteristics of all patients.

	IBS-D (*n* = 63)	IBS-C (*n* = 29)	IBS-M (*n* = 32)	Patients (*n* = 124), mean ± SD
Male sex	26	55	46	68 (54.8%)
Age (years)	12.51 ± 2.10	11.86 ± 1.68	12.25 ± 1.70	12.29 ± 1.92
Weight (kg)	48.87 ± 29.59	39.66 ± 32.15	39.44 ± 33.61	43.03 ± 13.54
Height (cm)	152.70 ± 13.76	148.00 ± 14.03	154.09 ± 13.16	151.96 ± 13.75
BMI (kg/m^2^)	18.71 ± 3.21	17.31 ± 2.95	18.32 ± 4.81	18.28 ± 3.65
25-OHD level (ng/mL)	15.25 ± 7.25	18.47 ± 7.37	16.22 ± 3.29	16.25 ± 6.58

BMI, body mass index; IBS-C, IBS with constipation; IBS-D, IBS with diarrhea; IBS-M, IBS with constipation and diarrhea; 25-OHD, 25-hydroxyvitamin-D; SD, standard deviation.

**Table 2 jcm-07-00500-t002:** Association of vitamin deficiency and sufficiency with abdominal pain and school absence.

		Vitamin D Deficiency Group (<20 ng/mL)	Vitamin D Sufficiency Group (≥20 ng/mL)
Abdominal pain	Daily (*n* = 38)	30 (78.9%)	8 (21.1%)
School absence	School absence because of IBS symptoms (*n* = 19)	18 (94.7%)	1 (5.3%)

IBS, irritable bowel syndrome.

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
