# Peer review of "Association of the Vitamin D Level and Quality of School Life in Adolescents with Irritable Bowel Syndrome"

_jcm, 2018, doi:10.3390/jcm7120500_

Round 1
Reviewer 1 Report
The paper seems well written, it is of modest originality.
at first a non-validated quality of life questionnaire was used. it would have been necessary to evaluate the data with a questionnaire already used in other studies together with the bristol scale.
Furthermore , in order to explain the pathophysiological mechanism, quantitative dosage of fecal calprotectin could be performed
Author Response
Point 1: At first a non-validated quality of life questionnaire was used. It would have been necessary to evaluate the data with a questionnaire already used in other studies together with the Bristol scale.
Response 1: I appreciate your comments and have revised the relevant details.(Page line )
As there is no official validated questionnaire, it is more difficult to evaluate pediatric or adolescent quality of life than that of adults. Therefore, the questionnaire used in this study is created based on the questionnaire used in studies on adults [1-3]. Moreover, adolescent quality of life is closely associated with school life as adolescents spend most of their time at school.
Point 2: Furthermore, in order to explain the pathophysiological mechanism, quantitative dosage of fecal calprotectin could be performed.
Response 2: I agree with your opinion and added information on fecal calprotectin in the Discussion section. The study on the relationship of irritable bowel syndrome symptoms and fecal calprotectin level is ongoing.
Fecal calprotectin is an objective and non-invasive sensitive inflammatory marker related to various pathological conditions. Therefore, further evaluation of the relationship between fecal calprotectin and irritable bowel syndrome symptom severity, especially quality of life, is needed.
1. Patrick, D.L.; Drossman, D.A.; Frederick, I.O.; DiCesare, J.; Puder, K.L. Quality of life in persons with irritable bowel syndrome: development and validation of a new measure. Dig Dis Sci 1998, 43, 400-411.
2. Tazzyman, S.; Richards, N.; Trueman, A.R.; Evans, A.L.; Grant, V.A.; Garaiova, I.; Plummer, S.F.; Williams, E.A.; Corfe, B.M. Vitamin D associates with improved quality of life in participants with irritable bowel syndrome: outcomes from a pilot trial. BMJ Open Gastroenterol 2015, 2, e000052, doi:10.1136/bmjgast-2015-000052.
3. Bengtsson, M.; Hammar, O.; Mandl, T.; Ohlsson, B. Evaluation of gastrointestinal symptoms in different patient groups using the visual analogue scale for irritable bowel syndrome (VAS-IBS). BMC Gastroenterol 2011, 11, 122, doi:10.1186/1471-230X-11-122.

Reviewer 2 Report
I've read with attention the paper of Cho et al. The methodology applied is overall correct, the results are reliable and adequately discussed. I've only some minor comments:
- The limitation of the study have to be listed and shortly discussed
- The quality of figures should be largely improved.
- The text requires an attentive revision by a native English speaker because of a number of typos and small mistakes.
Author Response
Point 1: The limitation of the study have to be listed and shortly discussed.
Response 1: I appreciate your insightful comments. I have mentioned the limitation of the study in the Discussion section.
This study is limited by the fact that vitamin D balance is dependent on numerous physiological and social factors such as exposure to sunlight, physical activity, and dietary habits. Another limitation is that the questionnaire used in this study is not universally validated. As there is no official validated questionnaire, it is more difficult to evaluate pediatric or adolescent quality of life than that of adults. Therefore, the questionnaire used in this study is created based on the questionnaire used in a study on adults [1-3]. Moreover, adolescent quality of life is closely associated with school life as adolescents spend most of their time at school.
Point 2: The quality of figures should be largely improved.
Response 2: Thank you for your advice. Higher quality figures have been provided now.
Point 3: The text required an attentive revision by a native English speaker because of a number of typos and small mistakes.
Response 2: Thank you for your advice and I have attached the relevant certification for MDPI English editing.
1. Patrick, D.L.; Drossman, D.A.; Frederick, I.O.; DiCesare, J.; Puder, K.L. Quality of life in persons with irritable bowel syndrome: development and validation of a new measure. Dig Dis Sci 1998, 43, 400-411.
2. Tazzyman, S.; Richards, N.; Trueman, A.R.; Evans, A.L.; Grant, V.A.; Garaiova, I.; Plummer, S.F.; Williams, E.A.; Corfe, B.M. Vitamin D associates with improved quality of life in participants with irritable bowel syndrome: outcomes from a pilot trial. BMJ Open Gastroenterol 2015, 2, e000052, doi:10.1136/bmjgast-2015-000052.
3. Bengtsson, M.; Hammar, O.; Mandl, T.; Ohlsson, B. Evaluation of gastrointestinal symptoms in different patient groups using the visual analogue scale for irritable bowel syndrome (VAS-IBS). BMC Gastroenterol 2011, 11, 122, doi:10.1186/1471-230X-11-122.
